# Insights into the Machine Learning Predictions of the Optical Response of Plasmon@Semiconductor Core-Shell Nanocylinders

Ehsan Vahidzadeh * and Karthik Shankar

Department of Electrical and Computer Engineering, University of Alberta, 9211-116 St.,
Edmonton, AB T6G 1H9, Canada
* Correspondence: vahidzad@ualberta.ca

**Abstract:** The application domain of deep learning (DL) has been extended into the realm of nanomaterials, photochemistry, and optoelectronics research. Here, we used the combination of a computer vision technique, namely convolutional neural network (CNN), with multilayer perceptron (MLP) to obtain the far-field optical response at normal incidence (along cylinder axis) of concentric cylindrical plasmonic metastructures such as nanorods and nanotubes. Nanotubes of Si, Ge, and $TiO_2$ coated on either their inner wall or both their inner and outer walls with a plasmonic noble metal (Au or Ag) were thus modeled. A combination of a CNN and MLP was designed to accept the cross-sectional images of cylindrical plasmonic core-shell nanomaterials as input and rapidly generate their optical response. In addition, we addressed an issue related to DL methods, namely explainability. We probed deeper into these networks' architecture to explain how the optimized network could predict the final results. Our results suggest that the DL network learns the underlying physics governing the optical response of plasmonic core-shell nanocylinders, which in turn builds trust in the use of DL methods in materials science and optoelectronics.

**Keywords:** energy; sensing; photocatalysis; in-silico design; classification; optimization; light-matter interactions; Maxwells equations; optical characterization; plasmonic hot carrier devices

## 1. Introduction

There exists a relatively small family of materials that exhibit localized surface plasmon resonances (LSPR) at visible and near-infrared wavelengths in their nanostructured forms [1]. This family includes Ag, Au, Cu, Al, HfN, ZrN, TiN, TaN, $Cu_2S$, and a few other compounds [1–4]. There has been considerable progress in tuning the shape of plasmonic nanomaterials. While plasmonic nanospheres are ubiquitous in nanoscience and nanotechnology, other shapes such as nanoprisms, nanocubes, nanoshells, and nanorods are becoming increasingly common [5,6]. Plasmonic nanomaterials are particularly interesting for applications in sensing, imaging, and catalysis where they can produce dramatic performance improvements due to the local electromagnetic field enhancement effect at metal-dielectric interfaces [7–10]. Plasmonic metamaterials allow the engineering of the photonic density of states and thus enable control over spontaneous and stimulated emission [11,12]. The hot carriers produced by plasmon decay can be used to drive chemical reactions [13–16]. Nearly all of the aforementioned applications benefit from the presence of a thin dielectric or semiconductor shell surrounding the plasmonic nanoparticles which allows for tuning of the LSPR peak and Q-factor, and enables extraction of hot carriers before thermalization and/or recombination [17]. Such a shell also confers photochemical and mechanical protection for the plasmonic core. In several cases, hot carrier-mediated oxidation or reduction is used to create the semiconductor or dielectric shell around the coinage metal core [18,19]. Time-consuming and resource-intensive empirical trials and/or computationally costly electromagnetic simulations are currently needed to obtain the optical properties of core-shell plasmon-dielectric metastructures for given values of material

type, shape, size, shell thickness, etc. This problem is particularly severe for cylindrical core-shell nanomaterials since exact analytical solutions do not exist, unlike the Mie theory analytical solutions available for nanospheres [20,21]. The Rayleigh–Gans approach of approximating nanorods with oblate spheroids fails to capture several features of interest in annular nanocylinders composed of core@shell and core@shell1@shell2 geometries. If an empirical approach is used to extensively document the optical absorption of cylindrical core-shell plasmon@dielectric metastructures to form a vast database, such a database composed of discrete values of various geometric parameters might still miss critical optical absorption features for specific geometries. The experimental task of obtaining the far-field optical response consists of two components, namely nanomaterials synthesis and nanomaterials characterization. The challenges in nanomaterials synthesis involve the difficulty in growing monodisperse core-shell nanocylinders and the sheer volume of experiments and human hours needed to synthesize core-shell nanoparticles of different sizes and material types to build a comprehensive library of structure-property relations. The optical characterization is also challenging and is not as simple as running the samples through a UV-Vis spectrophotometer. This is because spectrophotometers commonly measure the extinction of colloidal nanoparticles (in transmission mode) as a function of wavelength while determination of the optical absorption also requires accurate measurement of reflection, which is complicated for nanocylinders due to strong scattering and optical anisotropy. The goal of the present work is to use a machine learning-based approach to accurately predict the optical absorption of core-shell cylindrical plasmonic metamaterials for arbitrary geometrical and material parameters within a specified range of values. There are three key innovations in this work. First is the unusual nanotube and nanorod morphology of the investigated plasmonic noble metal-semiconductor heterojunctions. A second innovation consists of incorporating elements of computer vision to directly recognize the morphological parameters from the cross-sectional profile. Thirdly, we attempted to go beyond the blackbox nature of machine learning predictions by using the Shapley additive explanations framework to gain insights into the manner in which the artificial neural network (ANN) arrives at the results.

Finite-difference time-domain (FDTD) simulation is a powerful technique for estimating the far-field and near-field optical properties of metamaterials. Like every other finite element method, FDTD is based on solving partial differential equations (PDEs) on tiny building blocks of the defined structure. In FDTD, the PDEs are Maxwell's equations, and the small building blocks of the desired structure are called the Yee grid [22–24]. Solving Maxwell's equations on the Yee grid yields valuable information about the interaction of the electromagnetic wave with matter; this is an extensive area of study with applications in optoelectronics, photonics, photochemistry and sensing [25]. While there is no doubt about FDTD simulations' ability to model light/matter interactions, this method has its limitations. In general, the accuracy of finite difference methods depends on the mesh size, making the direct implementation of these methods extremely computationally expensive for nanomaterials. More accurate and reliable results are obtained only by using very fine mesh sizes [22,24–28]. Some techniques can be used to speed up the convergence of FDTD simulations and reduce the computational cost of FDTD simulations, such as exploiting the symmetry of the structure by applying symmetrical or anti-symmetrical boundary conditions, which can reduce the simulation cost by a maximum factor of eight [29] and dimensionality reduction where a 3D structure can be conceived as 2D or even 1D when the structure is infinite and homogeneous in one or two directions of the coordinate system [30,31]. Figure 1 exhibits a simplified representation of a 2D FDTD method. First, the 3D topology of a cylindrical structure that extends into infinity in the Z direction (along cylinder axis) is defined. Second the geometry is simplified into a corresponding 2D structure. Third, meshing is performed and the subsequent iterative solving of Maxwell's equations on this 2D structure yields the far-field absorption spectrum.

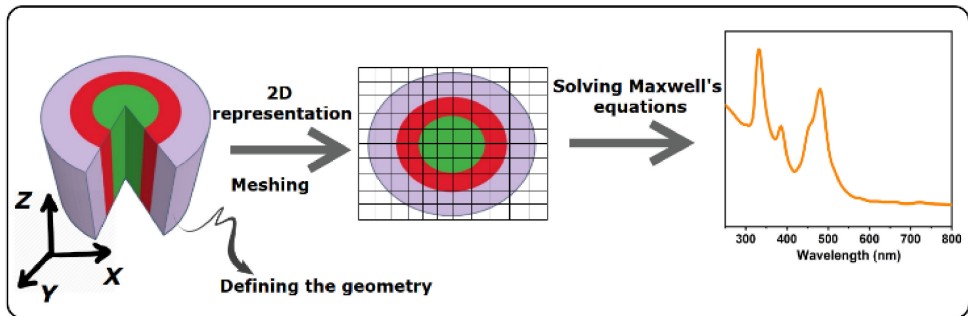

**Figure 1.** Schematic representation of the FDTD simulation—a user-defined 3D structure that goes to infinity in the Z direction is introduced. The FDTD engine simplifies the 3D structure into a 2D structure, splits it into small meshes, and solves Maxwell's equations on these meshes iteratively to then obtain the far-field optical response of the structure.

While FDTD simulation is known to be the most reliable method to simulate light-matter interactions, replacing FDTD with more rapid approaches has recently attracted a tremendous amount of attention [24,25,27]. The advent of artificial intelligence and deep learning (DL) has increased the efficiency of computer-aided processes in a wide range of research fields. The impact of DL on our lives is steadily increasing. DL has a wide range of applications, from speech recognition in smart home devices to face detection in phones [32]. DL algorithms rely on stacking a vast amount of data, training a model on the available data, and testing the developed model's prediction ability on data it has never seen before. Applying DL algorithms to replace sluggish and more computationally expensive FDTD methods was investigated recently. Compared with iterative processes such as FDTD methods, the computational cost of data-driven approaches such as DL methods occurs only during the model's training process. After the model is trained, obtaining the result is almost instantaneous with almost no computational cost [33]. In the broader field of nanophotonics, machine learning has been used to design photonic topological insulators [34], classify disordered Pt nanoparticles [35], design high-performance plasmonic nanosensors [36], and to enhance electron energy loss spectroscopy (EELS) for the fundamental study of nanoplasmonic phenomena [37]. Predicting the far-field [28,31,38] and near-field [32] optical properties of metastructures, and addressing the inverse design problem where an artificial neural network (ANN) suggests the optimal design parameters to obtain a desired optical response [28,32,33,39–42] are among the problems tackled by researchers thus far in the specialized area of plasmonic metastructures. While the end goal of replacing the slow and computationally expensive FDTD with a fast data-driven model developed by DL is intriguing, most of the attempts to date are based on introducing a set of physical parameters to ANN and acquiring the optical response from it [31–33,38–40]. This mimics the FDTD simulation, in which a structure's geometrical shape is given to the simulation engine as an input. Recently, convolutional neural networks (CNNs) were used to link the optical response of the structures with their cross-sectional images [31,43].

In this article, we demonstrated that a 2D image of a cylindrically shaped metamaterial can serve as an input to a convolutional neural network to obtain the desired optical response of a metamaterial. A CNN in combination with an MLP is used to instantly generate the absorption spectrum. While image classification and labeling are the primary usage cases of CNN thus far [31], herein the CNN was utilized to obtain both the categorical (type of material) and structural (radius and thicknesses of layers) parameters at the same time. An artificial neural network further used the physical parameters obtained with CNN to generate the desired optical spectrum with high accuracy. The most important contribution of this article is the development of a methodology to explain the results predicted by machine learning. By interpreting the MLP's output, we demonstrate that underneath the hood, the MLP follows a set of logical rules to predict the optical response of the metastructures that are explainable by the physics of light/matter interactions.

## 2. Materials and Methods

### 2.1. FDTD Simulations

Plasmonic metamaterials are artificially engineered materials that exhibit unique optical responses that do not occur in nature [44–46]. As a result, the optical properties of plasmonic metamaterials have been investigated extensively with possible applications in photochemistry, optics, optoelectronics, sensing, and nanophotonics [10,21,38,45]. In this work, cylindrically shaped core-shell metamaterials with a maximum of 3 layers were chosen as a case study. Core materials were selected from two important plasmonic metals (Ag and Au), and the shell material was chosen among the three ubiquitously used semiconductors (Si, Ge, and $TiO_2$). For the 3-layer structures, the outermost shell was again selected from either Ag or Au. In the real world, such concentric cylinders are encountered when arrays of vertically oriented semiconductor nanotubes are coated with a plasmonic absorber on both their inner and outer walls by a conformal deposition technique [2,47]. The polarization of the electric field was assumed along the *x*-axis. The radius of the cylinders and thickness of each layer was varied; the length of the cylinders was set to infinity along the *z*-axis so that the 3D structure can be simulated in a 2D environment. Table 1 summarizes the choices of the materials with the range of the radii used for FDTD simulations.

**Table 1.** Physical properties of the FDTD simulated metamaterials.

| Number of Layers | Core Materials | 1st Layer Materials | 2nd Layer Materials | Core Radii Range (nm) | 1st Layer's Thickness Range (nm) | 2nd Layer's Thickness Range (nm) |
|---|---|---|---|---|---|---|
| 1 | Au, Ag | NA | NA | 5–75 | NA | NA |
| 2 | Au, Ag | Si, Ge, $TiO_2$ | NA | 5–40 | 5–40 | NA |
| 3 | Au, Ag | Si, Ge, $TiO_2$ | Au, Ag | 5–25 | 5–25 | 5–25 |

The FDTD simulations were conducted using the commercially available Lumerical software package which solves Maxwell's equations on a Cartesian mesh for 2D or 3D structures. Lumerical provides a user-friendly graphical user interphase (GUI) where materials with different permittivities, shapes, and geometric configurations can be defined. Their far-field and near-field optical properties, including absorbance, reflectance, transmittance, electric field distribution, and Poynting vectors, can be simulated. Furthermore, Lumerical supports integrated scripting commands that help the user automate the whole simulation process when many simulations need to be performed. The sequence of simulations in Lumerical starts with defining the simulation environment, including the structure of the metamaterial, followed by the addition of the light source and monitors, and subsequent meshing. In the last step, Lumerical solves Maxwell's equations on the defined structure until it reaches convergence [48].

Concentric cylindrical structures with radii ranges mentioned in Table 1 were defined in the simulation environment. Since the cylinders were considered to extend to infinity in the Z direction, the simulation environment was set to 2D in X-Y Cartesian coordinates. A total-field scattered field (TFSF) [49] which ranges from 250–800 nm was selected as the light source. Lumerical's built-in material database provided the refractive index data for Ag, Au, Si, and Ge; the refractive index of $TiO_2$ was extracted and imported as a new material into Lumerical's database using the refractive index data reported in the literature [50]. The refractive index of the environment around the metamaterials was adjusted to air by setting its refractive index to 1. The metamaterials' light absorption was calculated using an absorption cross-section monitor surrounding the metamaterial. The number of frequencies for calculating the absorption properties was set to 200. A fine mesh of 2 nm was introduced to the simulation environment to ensure that the simulations converge. Meshing order was introduced so that when the cylinders overlap, the software automatically considers the inner cylinder's meshing. In total, 2426 different simulations were conducted; these

simulations were conducted automatically using Lumerical's native programming scripts, and the obtained spectra were outputted as a text file. A refractive index monitor showing a snapshot of the metastructure index's cross-section was placed inside the simulation environment. The refractive index monitor's output snapshots were further fed to the CNN neural network as the input images.

### 2.2. CNN and MLP Architectures and Implementations

Keeping in mind that our end goal was to obtain the absorption spectrum of a specific plasmonic metamaterial from a 2D image of its cross-section, the cross-sectional images obtained by Lumerical were used as the input, and the obtained absorption spectra were served as the output of the deep learning model. To link a 2D image to a spectrum, we divided the task into two different networks. The first one, which takes the 2D image as the input, is a CNN with the duty of realizing the physical properties such as materials used in each of the layers, the radius of the core, and the thickness of each shell. We named this CNN the physical properties recognizer network (*PPRN*). The second network was an MLP which accepts the physical properties generated by *PPRN* as the input and outputs the absorption spectra; this network is named as the spectrum generator network (SGN). Combining the *PPRN* and the SGN is a hybrid network that accepts a 2D image as input and outputs an absorption spectrum. While CNN is mainly used for classification problems, the *PPRN* network used here should successfully predict both continuous values (i.e., the radius of the core and thicknesses of the shell) and categorical values (i.e., type of materials used as core and shell). Prediction of continuous values falls within the regression problem territory where the mean absolute error (*MAE*) (Equation (1)) is the standard loss function. Additionally, predicting the categorical values is a multilabel classification problem where binary cross-entropy (*BCE*) (Equation (2)) is the expected loss function. In these two loss functions, $y_{predict}$ is the predicted value by the neural network and $y_{actual}$ is the reference value, and $\sigma$ is the sigmoid function. Since *PPRN* should predict a hybrid of categorical and continuous values, its loss function was set to a combination of *MAE* and *BCE* (Equation (3)), where $\alpha$ is a value between 0 and 1 which was optimized to reach the best result during the training process.

$$MAE = \frac{1}{n} \sum_n \left| y_{predict} - y_{actual} \right| \tag{1}$$

$$BCE = - \left[ y_{actual} \log \sigma \left( y_{predict} \right) + \left( 1 - y_{actual} \right) \log \left( 1 - \sigma \left( y_{predict} \right) \right) \right] \tag{2}$$

$$PPRN_{loss} = \alpha.MAE + (1 - \alpha).BCE \tag{3}$$

Figure 2 shows the combined architecture of the *PPRN* and SGN. The output shape of each of the layers is shown in the figure. The *PPRN*'s input was a series of images with the size of (107,82) in each red, green, and blue (RGB) channel, which caused the total dimension of each input image to be (107,82,3). The RGB values were divided by 255 (which is the maximum possible value for RGB filters) to facilitate the convergence process. The *PPRN* consisted of several convolutional layers followed by max-pooling, dropout, and flatten layers. During the simulation, the number of materials selected for the core, first layer, and second layer was chosen to be 2, 3, and 2, respectively, so the output of the *PPRN* was selected to consist of seven categorical values. In addition to that, the *PPRN*'s output consists of 3 continuous values related to the core radius, the thickness of the first layer, and the thickness of the second layer, making the output of *PPRN* consist of 10 different values in total. After the second max-pooling layer of the *PPRN*, the network was split into two branches. The left branch is related to the regression problem that outputs three continuous values with a linear activation function (core radius, first layer thickness, and the second layer's thickness). The branch on the right, outputs seven categorical values with a sigmoid activation function (core material, first layer's material, and second layer's material), and the combination of the output of these two branches with a shape of (1, 10) was used in the

loss function of the *PPRN* (Equation (3)). The total number of hidden layers (regardless of the last two dense layers, which was the network's output) was thirteen, making the total number of trainable parameters in *PPRN* equal 149,834. The activation function for each of the hidden layers was set to relu, and Adam [51] was chosen as the optimizer.

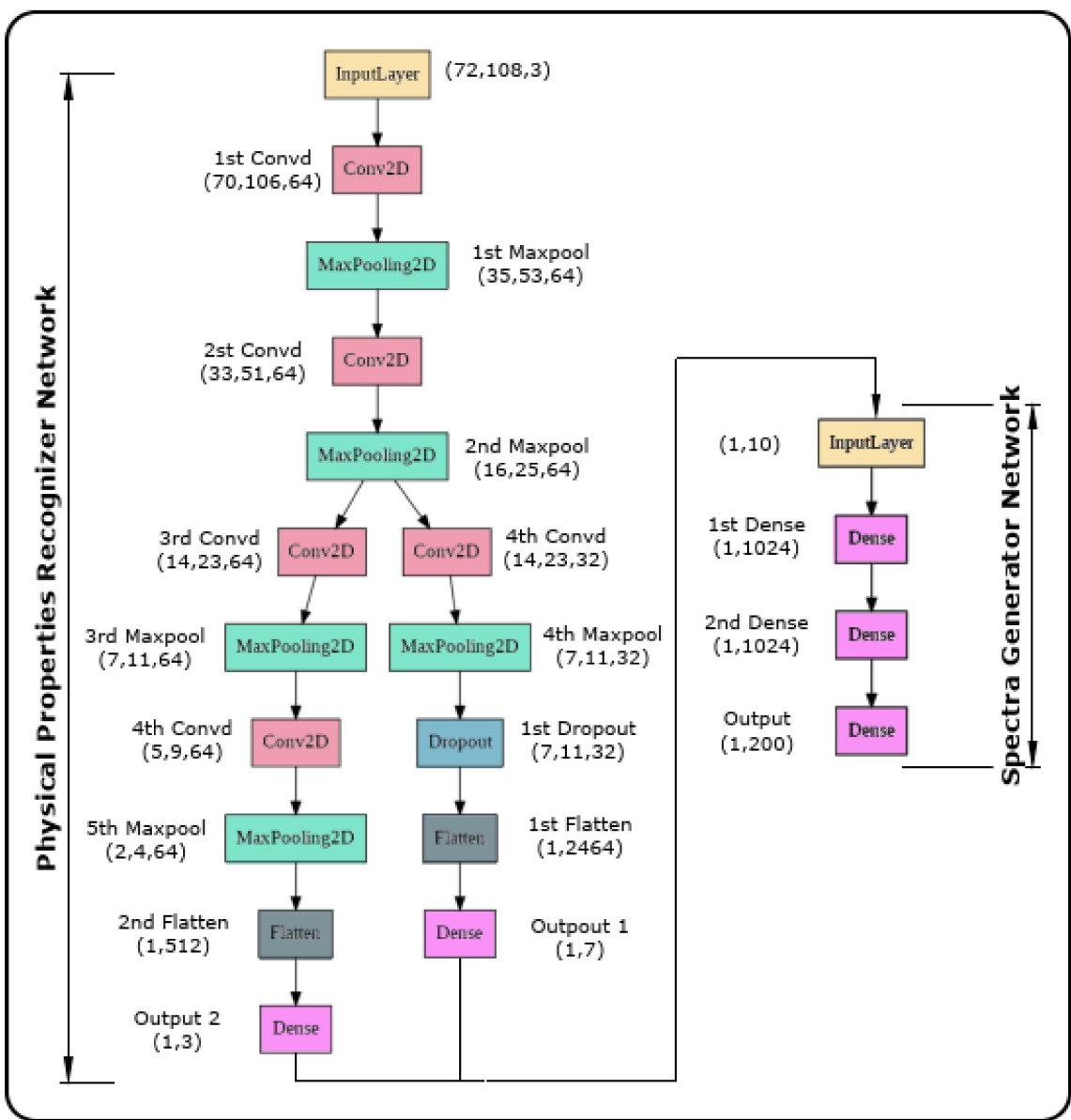

**Figure 2.** Schematic representation of the *PPRN* and SGN. *PPRN* is a CNN composed of several convolutional, max-pooling, dropout, flatten, and dense layers. SGN is an MLP composed of consecutive dense layers which uses the physical parameters suggested by *PPRN* to generate the final spectra.

Table 2 summarizes the ten physical features generated by the *PPRN*; the SGN should receive these ten physical properties and output absorption spectra with 200 data points; this is a multioutput regression problem, so the loss function for SGN was set to *MAE* (Equation (1)). Figure 2 also exhibits the SGN's architecture; it consists of 2 hidden layers with 1024 nodes and output with 200 nodes. The total number of trainable parameters for this network was 1,265,864. The activation function of the hidden layers was set to relu, and a linear activation function was chosen for the last layer of this MLP. Like *PPRN*, Adam was selected as the optimizer of SGN. It is worthy of mention that the hyperparameters,

including the number of hidden layers, number of neurons in each layer, choice of activation function, optimizer, learning rate, and batch size, were optimized, and the architectures reported in Figure 2 are the optimized layouts.

**Table 2.** Physical properties outputted by *PPRN*/used by the SGN network as input.

| Categorical Features | | | | | | | Continuous Features | | |
|---|---|---|---|---|---|---|---|---|---|
| Core Material (Au) | Core Material (Ag) | Shell1 Material (TiO$_2$) | Shell1 Material (Si) | Shell1 Material (Ge) | Shell2 Material (Au) | Shell2 Material (Ag) | Core Radius | Shell1 Thickness | Shell2 Thickness |

## 3. Results and Discussion

### 3.1. FDTD Simulation Results

The simulation considered nanocylinders, which are infinite in length along the *x*-axis and oriented along the *z*-axis. Since the propagation direction of the electric field of light was also along the *z*-axis, the simulations captured the optical absorption behavior at normal incidence of an array of non-interacting infinitely long core-shell nanocylinders. Thus, only the transverse plasmon resonances of each nanocylinder were considered and the longitudinal resonances were ignored. Figure 3 exhibits randomly chosen structures within the set of annular core-shell nanocylinders and their corresponding absorption spectra; the inset of each image shows the snapshot of the cross-section of the cylindrical metamaterials. For instance, the bare gold and silver nanocylinders in Figure 3a–c show the characteristic LSPR peaks in air of nanostructured Au and Ag at ~510 nm and 370 nm respectively. The LSPR peak of Ag in Figure 3c is narrower and sharper than that of Au in Figure 3a,b because of the lower dielectric loss of Ag, leading to a smaller surface plasmon bandwidth. The higher permittivity (compared to air) of the concentric dielectric shell shifts the transverse plasmon resonance of the noble metal core to longer wavelengths (e.g., Figure 3d). For single-core, single-shell nanocylinders, additional excitonic peaks appeared to due to the size quantization of the semiconductor shell surrounding the nanocylinders (e.g., Figure 3e). Plexcitonic effects were also captured by the simulations. When an excitonic resonance was very close to the plasmon resonance, Rabi splitting due to strong coupling (e.g., Figure 3f,h) or anomalous peak broadening due to weak coupling (e.g., Figure 3g) were also seen [52]. For single-core, double-shell nanocylinders (annular dielectric cylinders coated on the interior and exterior with identical metals), additional collective modes appeared due to the interaction between the core- and shell resonances (Figure 3i–l).

The plasmon resonance of each layer in metamaterials can be engineered in such a way that plasmon hybridization occurs and the total absorption cross section overlaps at a specific wavelength, a phenomenon called super-absorption [46,53,54]. Super-absorption will lead to a strong light/matter interaction at a specific wavelength. This ability makes the superabsorbers desirable for optoelectronic applications such as photovoltaics, photodetectors, highly selective photocatalysts, surface-enhanced spectroscopy, and sensing [55]. A quick look at Figure 3 gives insight into the diversity of the optical response of plasmonic metamaterials simulated in this work. It is therefore clear that a careful design of the metamaterials' building blocks is needed to achieve the desired optical response. Furthermore, the simulated structures exhibited a variety of super-absorption responses at wavelengths ranging from 300 nm to 600 nm, which makes them ideal candidates for the aforementioned applications.

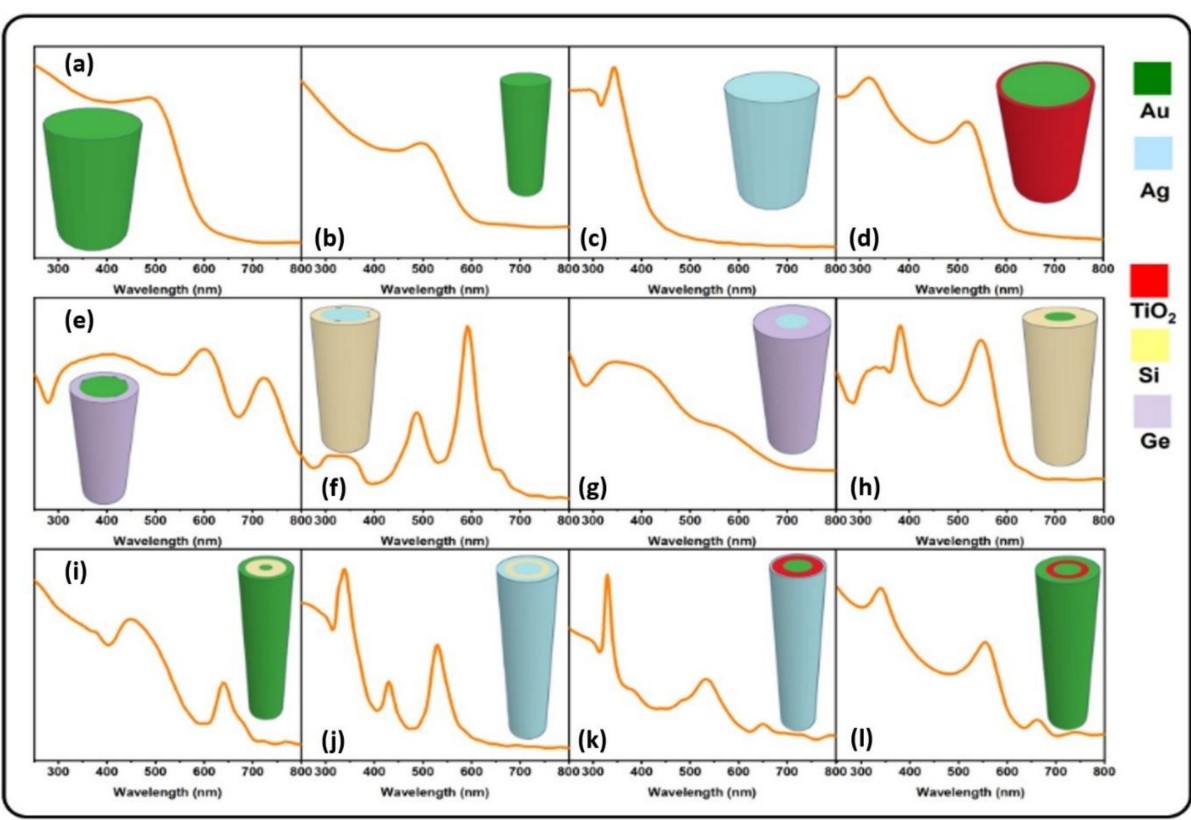

**Figure 3.** The simulated optical absorption spectra of randomly selected plasmonic metamaterials—each of the materials is represented by a specific color, which is shown on the right side of the figure.

### 3.2. Training of PPRN and SGN

The training process was conducted on a computer equipped with an Intel(R) Xeon(R) CPU and a Tesla T4 GPU. The neural network architecture was defined and trained using Tensorflow (v.2.4.1). For training the Physical Properties Recognizer Network (*PPRN*) and Spectrum Generation Network (SGN), the available data were split into training and test sets. Seventy percent of the data was used for the training, ten percent for validation, and the remaining twenty percent was allocated to test the trained models' ability to predict the unseen data. The maximum number of epochs for training was set to 1500, and a call-back function with patience of 10 on the validation loss was used to stop the training at a specific epoch to prevent overfitting. The training and validation loss as a function of the number of epochs for training the *PPRN* and SGN are shown in Figure 4. The *PPRN*'s loss (Figure 4a) exhibited a gradual decrease as the epoch increased and reached values as low as $10^{-3}$ for both the training and the validation loss. Since *PPRN* also classifies the input images, classification accuracy is an essential factor for judging its ability. Figure 4b exhibits the *PPRN*'s accuracy of classification; after only 178 epochs, the accuracy reached the excellent value of 0.99 for both training and validation, which implies the superior ability of *PPRN* to classify the input images into the correct categories. The decrease of loss as a function of the number of epochs for SGN's training is also evident in Figure 4c. After 720 epochs, the SGN's loss reached 0.013 and 0.019 for training and validation, respectively. The proximity of the training and validation losses of both *PPRN* and SGN suggests that the optimized architecture designed for these two networks can generalize and predict unseen data effectively, and overfitting did not occur for either of these two trained networks. Adding more complexity to these two networks (i.e., adding more hidden layers or changing the number of neurons in each hidden layer) resulted in a network that was incapable of generalization.

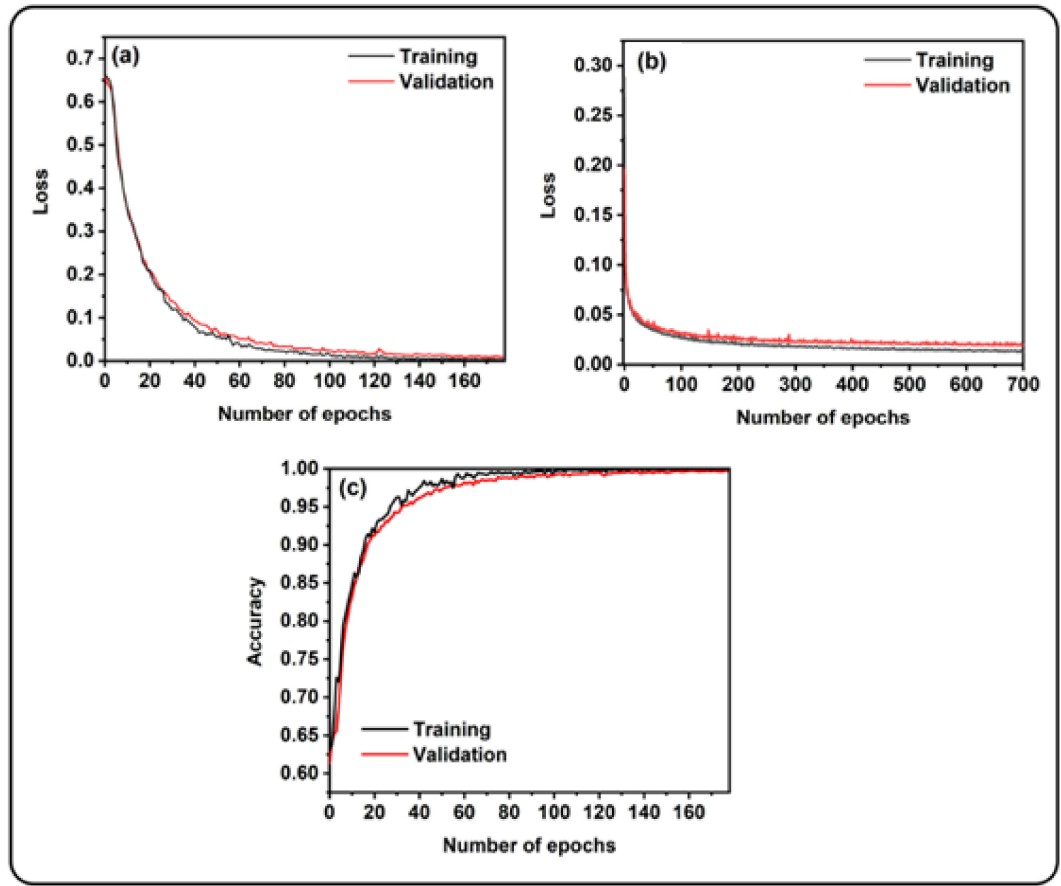

**Figure 4.** The training loss as a function of the number of epochs for (**a**) *PPRN* and (**b**) SGN networks and (**c**) accuracy as a function of the number of epochs for *PPRN*.

To further clarify how the arcitecture of the networks was optimized, the results of the validation errors for the SGN with different architectures are reported in Table 3. These results suggest that a model with fewer than three layers or a model with tanh as the activation function cannot perform predictions well on the validation set. In addition, when the complexity of the SGN exceeds its optimal structure (by increasing the number of layers or the number of neurons in each layer), the model tends to overfit and the validation error increases.

**Table 3.** The validation error for SGN networks with different architectures.

| SGN Architectures | Activation for the Hidden Layers | Validation Error |
| --- | --- | --- |
| {1024,1024,200} (used in this study) | relu | 0.019 |
| {1024,1024,200} | tanh | 0.034 |
| {1024,1024,500,200} | relu | 0.025 |
| {512,200} | relu | 0.055 |

### 3.3. Comparison between the Results Obtained by PPRN and SGN, with FDTD Simulations

After the training process was completed, the *PPRN* and SGN models were further used to calculate the test samples' absorption spectra (samples on which the model was not trained). Figure 5 exhibits a few randomly selected examples from the test set, the FDTD simulated absorption spectrum, and the predicted spectrum generated by the *PPRN* and SGN. As is evident from these results, the spectra generated by the combination of *PPRN* and SGN correlate well with the FDTD simulated spectra. Because the FDTD simulation's runtime to create the absorption spectrum was on average in the range of 100 s and the

time elapsed for *PPRN* + SGN to generate similar spectra was on average around 0.1 s, the *PPRN* + SGN could deliver the same task 1000 times faster than FDTD simulation.

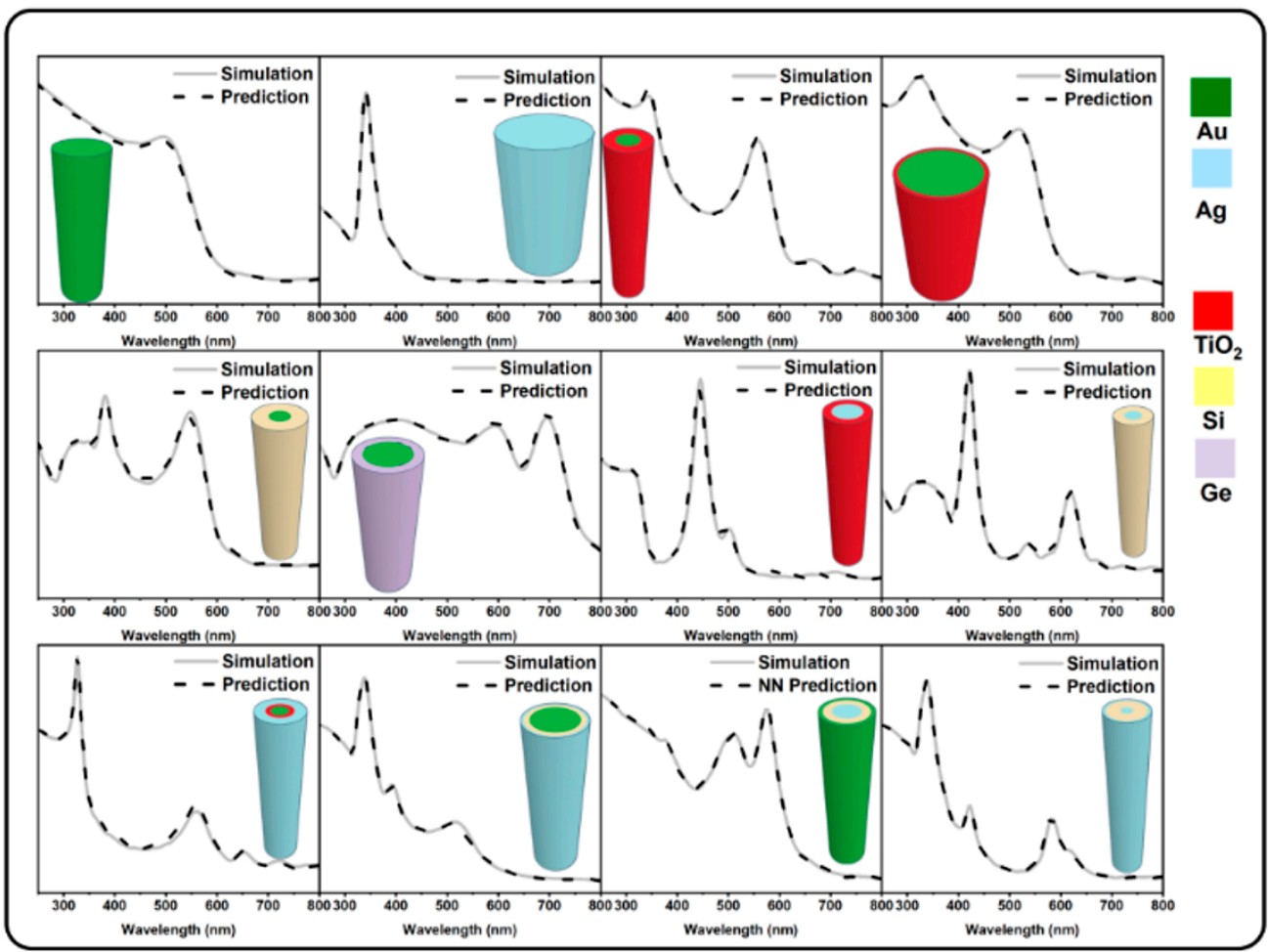

**Figure 5.** The comparison between the absorption spectrum obtained by FDTD simulations and final models (developed by DL) for a few randomly selected structures from the test set—these results prove that the final model developed by DL can swiftly yield the same spectrum with high accuracy.

### 3.4. Interpreting the PPRN and SGN

To further understand how the *PPRN* model interprets the input images to extract the features and physical properties out of the input image, each hidden layer's outputs were plotted as the model progresses. Figure 6 exhibits the input image fed to the *PPRN* and outputs of randomly selected filters in each of the layers embedded in *PPRN*. As is evident from this figure, the *PPRN* model can extract features such as edges, colors, and outlines. All these features extracted by the *PPRN* model help it decide the physical properties of the input image. These results showcase the outstanding ability of *PPRN* for the interpretation of different physical properties of the input image by extracting a comprehensive set of details out of it. These results, alongside the 99 percent accuracy for classification and losses as low as $10^{-3}$ for training and validation, guarantee that *PPRN* is a robust network for deciphering the physical properties from the input image.

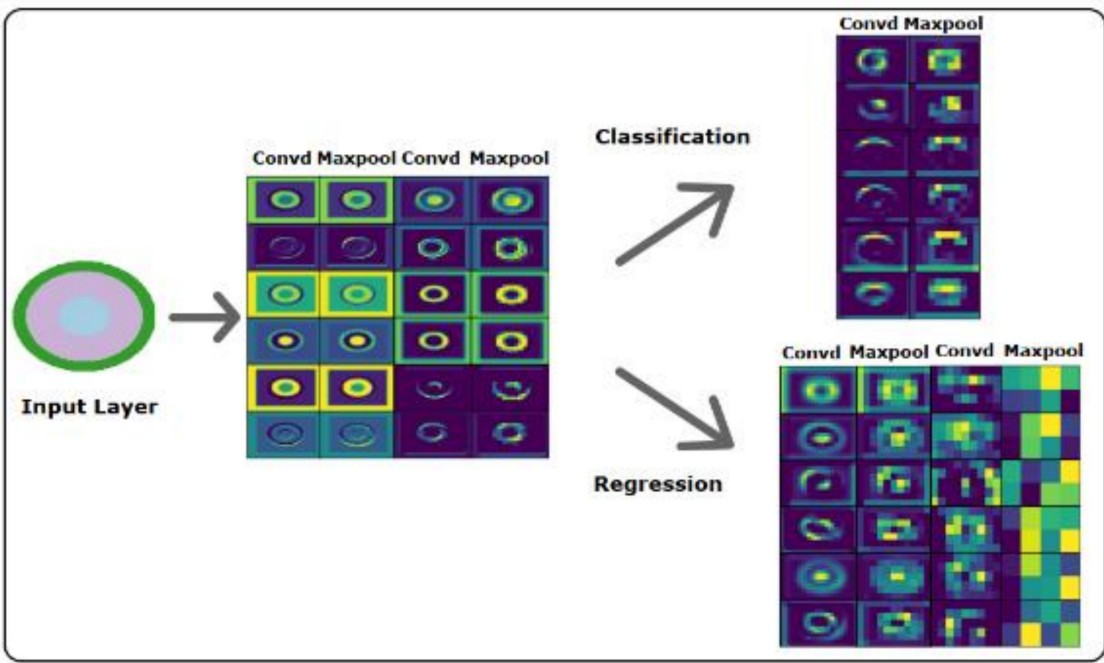

**Figure 6.** The output of randomly selected layers of *PPRN*. The heterogeneity of the extracted features in the output of each layer shows the *PPRN*'s caliber in finding physical properties.

The black-box nature of ANNs has made their interpretation a challenging task [56]. While human beings follow general wisdom for the decision making process, it is hard to understand how exactly an ANN decides to reach a specific prediction based on the input features [57,58]. Recently, a new framework for interpreting the behavior of ANNs was introduced which is called SHAP (Shapley Additive exPlanations) [58]. SHAP values are numbers assigned to each of the input features based on game theory. By removing the feature from the feature space and calculating to what degree the output was affected by this change, SHAP values are calculated [56]. SHAP values measure how the ANN gave importance to a feature for a particular prediction. A higher absolute SHAP value corresponds to the higher significance of that specific feature. Positive SHAP values mean that the feature was contributing to increasing the predicted value and vice versa. To understand each feature's effect on the prediction, the SHAP method was implemented to explain the SGN network's behavior. Figure 7a exhibits the mean calculated SHAP values for each of the features (these features are listed in Table 2) used by the SGN network. It is evident from Figure 7a that the continuous features (i.e., sizes of the materials and structures listed in Table 2) obtained higher SHAP values compared with the categorical values (i.e., materials used in each of the layers). What this means is that the output prediction of SGN is heavily affected by the size of the structure (continuous features) and the nature of the material. This is a coherent deduction since one of the photophysical properties of multilayer plasmonic metamaterials is the non-uniqueness of their optical response wherein the far-field optical response of different plasmonic metamaterials with the same size range resemble each other [59,60]. As a result, it is not surprising to see that more importance was given to continuous features by the SGN for decision-making.

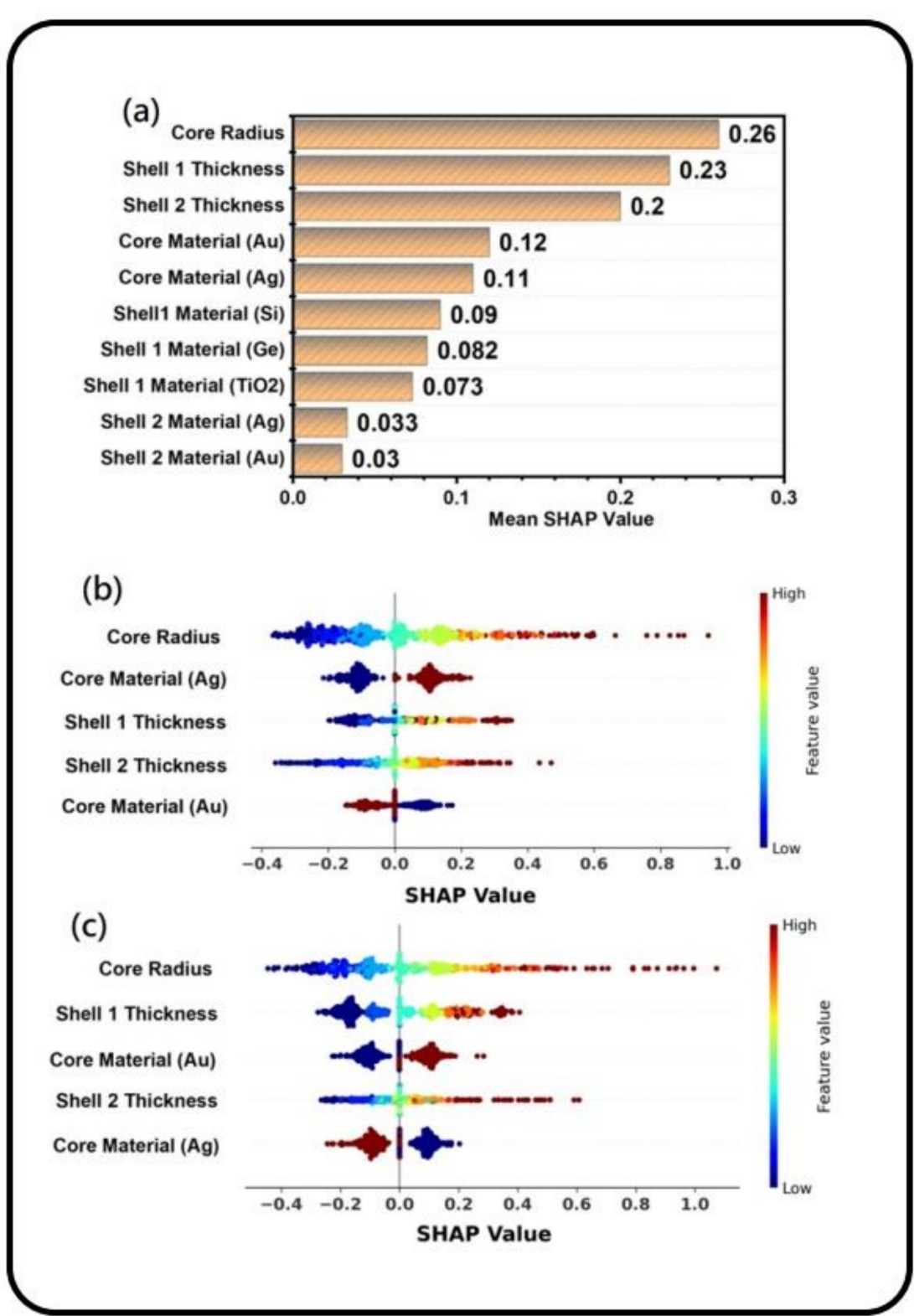

**Figure 7.** (**a**) Calculated importance of each of the features (absolute SHAP values) for SGN. (**b**) Calculated SHAP values of the SGN network corresponding to 400 nm. (**c**) Calculated SHAP values of the SGN network corresponding to 500 nm.

Since the output of the SGN network is 200 data points, each of which corresponds to the far-field absorption response of the plasmonic structures at a specific wavelength (ranging from 250 nm to 800 nm), SHAP values of the SGN for the top five most essential

features in a particular wavelength of interest were calculated; 400 nm and 500 nm were chosen as the wavelengths of interest as it is known that nanostructures of Ag (350–400 nm) and Au (500–550 nm) show strong interaction with incident light at these wavelengths, respectively [7,61–63] due to the LSPR phenomenon. Figure 7b shows the SHAP values of SGN corresponding to 400 nm. At this specific wavelength, Ag's importance as the type of the material used as a core material is high, even higher than continuous values such as shell 1 and shell 2 thicknesses. Besides, Ag's presence as the core material contributed positively to the light absorption at this specific wavelength, proving that the SGN network could pinpoint Ag's importance on the output optical response in this particular spectral region. A similar trend for Au can be seen in Figure 7c, where the SHAP values of SGN corresponding to 500 nm are shown. Overall, the interpretations obtained by the SHAP value calculations provide solid evidence that the SGN is acquiring an abstract understanding of the most significant physical characteristics that regulate the light/matter interaction. In other words, rather than randomly assigning numbers as an output, the SGN is deducing logical interpretations out of the features imported to this network. While still primitive, the SHAP value calculations are a step in the direction of reducing the black box nature of ANNs and improving the explainability of their predictions.

## 4. Conclusions

Plasmonic metamaterials exhibit a diverse optical response for visible light frequencies and are important materials for optoelectronic applications such as optical sensors, light emitting devices, photocatalysts and photovoltaics. Numerically solving Maxwell's equations on a finite element grid using FDTD simulation is an essential tool for understanding the optical properties of these materials to avoid time- and resource-intensive empirical examination of the entire available parameter space. FDTD is a computationally expensive and slack process. One of the best ways to improve the efficiency of a process is by increasing its speed without losing precision. As a result, replacing FDTD with swifter DL methods is a hot topic in optoelectronics research. Furthermore, with the ultimate goal of replacing FDTD with DL methods, there should be a framework to build trust in the models developed by DL. Herein, we demonstrated that more rapid models developed by DL can replace FDTD. A new network consisting of a combination of convolutional neural network (CNN) and multilayer perceptron (MLP) models was designed, which receives the cylindrical plasmonic core-shell nanomaterials' cross-sectional images as input and rapidly generates their absorption spectrum with outstanding precision. To understand how CNN and MLP are generating their predictions, the interpretation of the predictions of the CNN was made by visualizing each hidden layer's output, revealing how CNN could understand the outline and shape of the images which it receives as an input. Further interpretation of the MLP predictions was made using SHAP value calculations which opened a window into the otherwise black-box nature of the ANNs. SHAP feature importance calculations proved that the SGN is learning the physics governing the plasmonic metastructure/light interactions behind the scenes. Further progress in explaining the predictions of machine learning based approaches is needed, which might also provide new physical insights. These results demonstrate the viability and integrity of DL approaches as potential alternatives to both FDTD simulations and experiments.

**Author Contributions:** Conceptualization, E.V.; methodology, E.V.; software, E.V.; validation, E.V. and K.S.; formal analysis, E.V. and K.S.; investigation, E.V.; resources, K.S.; data curation, E.V.; writing—original draft preparation, E.V.; writing—review and editing, K.S.; visualization, E.V.; supervision, K.S.; project administration, K.S.; funding acquisition, K.S. All authors have read and agreed to the published version of the manuscript.

**Funding:** This research was funded by Future Energy Systems (FES) Canada First Research Excellence Fund, grant number T12-P02; the National Research Council Canada (NRC)-University of Alberta NanoInitiative, project A1-014009; and NSERC. The APC was funded by FES.

**Institutional Review Board Statement:** Not applicable.

**Informed Consent Statement:** Not applicable.

**Data Availability Statement:** The data presented in this study are openly available in [FigShare] at: https://doi.org/10.6084/m9.figshare.20224314.v1. The source codes for this work are publicly available at in [GitHub] at: https://github.com/vahidzad/FDTD-AIExplain.git.

**Conflicts of Interest:** The authors declare no conflict of interest.

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
