# Peer review of "Insights into the Machine Learning Predictions of the Optical Response of Plasmon@Semiconductor Core-Shell Nanocylinders"

_2673-7256, doi:10.3390/photochem3010010_

Round 1
Reviewer 1 Report
Deep learning-related innovations in physics and biology are increasingly important. The authors have made important and useful progress in this direction in the plasmonics and photonics domain, and this is a useful paper for the research community. However, their methodology choices are often too opaque or confusing to be understandable, especially for physical scientists who want to use the newly presented tools in their own research. Cross-validation seems to have been inadequately performed in the model training as well. Thus I recommend significant revisions before acceptance, as well as urge the authors to make their codes and some sample data publicly available upon publication.In my copy of the manuscript, many symbols seem to have been replaced with "@" as a conversion error. Please check this.
FDTD computation discussion on p. 2 - for a "standard" academic GPU, what is the rough time estimate of FDTD for a given particle's model to finish in a standard analysis? Some general order of magnitude may help give the readers an appreciation of the problem.
" A CNN in combination with an ANN is used to instantly generate the absorption spectrum "
Reading this statement concerns me, as a convolutional neural network is a type of artificial neural network so this sentence does not make sense. This is incorrect use of very basic jargon in AI/ML. "Artificial neural network" is one of the broadest possible terms that encompasses CNNs, RNNs, DNNs, etc. so this does not give enough details on the methods. What is the nature of this second ANN? Following this issue, the authors have overall not motivated or described their choice of neural network parameters and model architecture adequately, they just say "It is worthy of mentioning [typo] that the hyperparameters, including the number of hidden layers, number of neurons in each layer, choice of activation function, optimizer, learning rate, and batch size, were optimized, and the architectures reported in Figure 2 are the optimized layouts". What was the nature and methodology of this optimization? What other parameters/settings were tried and what were the criteria for rejecting them? Why was relu chosen over tanh or other alternatives for example? The methods need to be sufficiently detailed to guide other researchers who are interested in this technique of how to optimize and make good technical decisions in this domain. If it is too much information for the main text, an SI document should be uploaded with this information and clearly referred to in the main text for reproducibility and explainability.
Then, the current test/train validation method seems to just have one test/train division, rather than using more modern cross-validation methods which try many combinations of test/training on randomly chosen sampling (e.g. see https://www.sciencedirect.com/science/article/abs/pii/S0031320315000989 "Performance evaluation of classification algorithms by k-fold and leave-one-out cross validation"). My understanding is that just one test/train/validation division is no longer acceptable for establishing the performance of an artificial neural network model. Also what logic or analysis motivated an acceptable number of samples in this analysis? What is the minimal number of training examples (roughly) that other researchers need to use to use within this technique, at least in the nanophotonics/nanoplasmonics domain that these methods are being applied to?
"Overall, the interpretations obtained by the value calculations provide solid evidence that the SGN is learning the underlying physics governing the light/matter interaction."
This is a bit of an overstatement, something like "... is learning abstractions of the most important physical features governing the light/matter interactions in this context" is more accurate (you do not have to use my phrasing, just giving an example of how to soften this in a more accurate way).
Figure 7 especially is very cool, good job.
"Although ANNs are designed to mimic the human brain in decision-making" -- some specific neuro/bio-inspired ANNs are made to mimic some part of the human brain in neuroscience research, most ANNs especially in harder AI/ML domains are not other than in very abstract hand-wavy ways. The authors have not sufficiently argued that their neural network architectures are particularly bio-inspired, and this statement is not really relevant to the conclusions so I recommend softening this to avoid overstatement that may distract or annoy readers.
Last, my strong recommendation is to upload your neural network analysis codes and at least some sample data to a publicly available database immediately upon publication rather than the current "available upon request" statement. I can't think of any reason why you wouldn't be able to do this, as this is not based on confidential human data or anything like that, and you are advertising this method as a new standard that you want other groups to use. Especially given the possibly quite complex optimization process that was only vaguely hinted at in the main text, pubicly available codes seem necessary for interested technical readers to understand this manuscript.
Author Response
Please enclosed pdf file which contains the Response to Reviewer#1.

Reviewer 2 Report
Ehsan Vahidzadeh et al. investigated “Beyond the Blackbox: Explaining the Machine Learning Predictions of the Optical Response of Plasmon@Semiconductor Core-Shell Nanocylinders”. In my view, the article has some serious shortcomings. My comments are listed below.
(1) The title is quiet long and confusing. Please modify the title.
(2) There are many grammatical errors. Rectify them carefully.
(3) The abstract lacks important findings of the research. Rewrite it.
(4) The novelty of the work is not stated properly. Clearly write the novelty.
(5) Include the previous researches on proposed work and compare your results with them.
(6) Write the physics behind the nature of variation in the results obtained for fig. 2, 3
Author Response
Reviewer #2
The authors thank the referee for investing the time in reviewing our manuscript and providing us with constructive feedback.
(1) The title is quiet long and confusing. Please modify the title.
Response: We have changed the title to "Insights Into the Machine Learning Predictions of the Optical Response of Plasmon@Semiconductor Core-Shell Nanocylinders". I am afraid we cannot further shorten the title since it would then detract from the information which is necessary to convey in the title.
(2) There are many grammatical errors. Rectify them carefully.
Response: All grammatical errors have been fixed in the revision.
(3) The abstract lacks important findings of the research. Rewrite it.
Response: We deleted the second sentence of the abstract and added one new sentence describing the work performed and the results obtained. The modified abstract is as follows:
The application domain of deep learning (DL) has been extended into the realm of nanomaterials, photochemistry and optoelectronics research. Here we used the combination of a computer vision technique, namely convolutional neural network (CNN), with multilayer perceptron (MLP) to obtain the far-field optical response at normal incidence (along cylinder axis) of concentric cylindrical plasmonic metastructures such as nanorods and nanotubes. Nanotubes of Si, Ge and TiO2 coated on either their inner wall or both their inner and outer walls with a plasmonic noble metal (Au or Ag) were thus modeled. A combination of a CNN and MLP was designed to accept the cross-sectional images of cylindrical plasmonic core-shell nanomaterials as input and rapidly generate their optical response. In addition, we addressed an issue related to DL methods, namely explainability. We probed deeper into these networks' architecture to explain how the optimized network could predict the final results. Our results suggest that the DL network learns the underlying physics governing the optical response of plasmonic core-shell nanocylinders, which in turn, builds trust in the use of DL methods in materials science and optoelectronics.
(4) The novelty of the work is not stated properly. Clearly write the novelty.
Response: The novelty of the work is now explicitly stated in the final sentences of the first paragraph of the Introduction.
There are three key innovations in this work. First is the unusual nanotube and nanorod morphology of the investigated plasmonic noble metal-semiconductor heterojunctions. A second innovation consists of incorporating elements of computer vision to directly recognize the morphological parameters from the cross-sectional profile. Thirdly, we have attempted to go beyond the blackbox nature of machine learning predictions by using the Shapley additive explanations framework to gain insights into the manner in which the ANN arrives at the results.
(5) Include the previous researches on proposed work and compare your results with them.
Response: The novelty of our approach prevents us from complying with this suggestion.
- This is the first ever work that involves integrating or incorporating computer vision into the machine learning based predictions of the optical response of plasmonic nanomaterials
- This is also the first work to the best of our knowledge which uses machine learning to study core-shell plasmon@semiconductor nanorods and nanotubes. The two closes articles we found were on carbon nanotubes and gold nanocubes and therefore largely irrelevant to our work.(i) EM Khabushev et al. "Machine Learning for Tailoring Optoelectronic Properties of Single-Walled Carbon Nanotube Films", J. Phys. Chem. Lett. 2019, 10, 21, 6962–696 (ii) J. A. Arzola-Flores and A. L. González "Machine Learning for Predicting the Surface Plasmon Resonance of Perfect and Concave Gold Nanocubes", J. Phys. Chem. C 2020, 124, 46, 25447–25454.
(6) Write the physics behind the nature of variation in the results obtained for fig. 2, 3.
Response: We had already done this to the best of our ability in the previous version of the manuscript, see lines 273-287 reproduced below. Additional explanation and emphasis of the results in Figure 3 are outside the scope of the present work and would distract the reader from the focus on machine learning. predictions of the optical response.
Figure 3 exhibits randomly chosen structures within the set of annular core-shell nanocylinders and their corresponding absorption spectra; the inset of each image shows the snapshot of the cross-section of the cylindrical metamaterials. For instance, the bare gold and silver nanocylinders in Figure 3a-c show the characteristic LSPR peaks in air of nanostructured Au and Ag at ~510 nm and 370 nm respectively. The LSPR peak of Ag in Figure 3c is narrower and sharper than that of Au in Figures 3a and 3b because of the lower dielectric loss of Ag, leading to a smaller surface plasmon bandwidth. For single core, single shell nanocylinders additional excitonic peaks appear to due to the size quantization of the semiconductor shell surrounding the nanocylinders (e.g. Figure 3d). Plexcitonic effects are also captured by the simulations. When an excitonic resonance is very close to the plasmon resonance [52], Rabi splitting due to strong coupling (e.g. Figure 3f) or anomalous peak broadening due to weak coupling (e.g. Figure 3g) are also seen. For single core, double shell nanocylinders (annular dielectric cylinders coated on the interior and exterior with identical metals), additional collective modes appear due to the interaction between the core- and shell resonances (Figures 3i-l).
Reviewer 3 Report
In my opinion, the article will be of high interest for the Photochem audience. The research was performed at high level, with great attention to details and provides interesting and useful insight in the area of study. Therefore, the article definitely deserves to be accepted to the Photochem.
I can see the following points of strength in the manuscript:
the introduction is very detailed and provides good understanding of the subject of research,
the methodology is described well and allows for other researchers to reproduce the results correctly,
the description of results is very detailed,
the illustrative material (figures and tables) very well illustrates the results of the study,
the conclusions are very well supported by the study results.
Author Response
We are deeply grateful to Reviewer#3 for the time he/she invested in reviewing our work and for the enthusiastic endorsement of the manuscript.
Round 2
Reviewer 1 Report
The authors have sufficiently addressed the reviewer's concerns. Congratulations!